# Functional Characterization of Ubiquitination Genes in the Interaction of Soybean—*Heterodera glycines*

**DOI:** 10.3390/ijms231810771

**Published:** 2022-09-15

**Authors:** Nawei Qi, Jichen Yan, Piao Lei, Xiaofeng Zhu, Xiaoyu Liu, Yuanhu Xuan, Haiyan Fan, Yuanyuan Wang, Lijie Chen, Yuxi Duan

**Affiliations:** 1Nematology Institute of Northern China, Shenyang Agricultural University, Shenyang 110866, China; 2College of Plant Protection, Shenyang Agricultural University, Shenyang 110866, China; 3Institute of Plant Protection, Liaoning Academy of Agriculture Sciences, Shenyang 100161, China; 4College of Sciences, Shenyang Agricultural University, Shenyang 110866, China; 5College of Biological Science and Technology, Shenyang Agricultural University, Shenyang 110866, China

**Keywords:** ubiquitination, *GmPUBs*, soybean, *Heterodera glycines*, soybean cyst nematode

## Abstract

Ubiquitination is a kind of post-translational modification of proteins that plays an important role in plant response to biotic and abiotic stress. The response of soybean *GmPUB* genes to soybean cyst nematode (SCN, *Heterodera glycines*) infection is largely unknown. In this study, quantitative real-time PCR (qRT-PCR) was performed to detect the relative expression of 49 *GmPUB* genes in susceptible cultivar William 82 and resistant cultivar Huipizhi after SCN inoculation. The results show that *GmPUB* genes responded to cyst nematode infection at 1 day post-inoculation (dpi), 5 dpi, 10 dpi and 15 dpi. The expression levels of *GmPUB16A*, *GmPUB20A*, *GmCHIPA*, *GmPUB33A*, *GmPUB23A* and *GmPUB24A* were dramatically changed during SCN infection. Furthermore, functional analysis of these *GmPUB* genes by overexpression and RNAi showed that *GmPUB20A*, *GmPUB33A* and *GmPUB24A* negatively regulated soybean resistance under SCN stress. The results from our present study provide insights into the complicated molecular mechanism of the interaction between soybean and SCN.

## 1. Introduction

Ubiquitination is involved in many biological processes, including the cell cycle, hormone signaling, signal transduction, photomorphogenesis and senescence in plants. Ubiquitination modification participates in a series of reactions of the ubiquitin-activating enzyme (E1), ubiquitin-conjugating enzyme (E2) and ubiquitin-ligase enzyme (E3) [1]. Polyubiquitin-modified target proteins are generally degraded by the 26S proteasome, called the ubiquitin-26S proteasome system (UPS) [2]. The UPS is responsible for selecting, targeting and proteolyzing specific degradation substrates. Its components are particularly abundant in plants and are key hubs for regulating various plant cell processes. Hence, the UPS is essential for development and stress response in plants [1,3].

Ubiquitin ligase E3 is classified into two groups: the HECT domain family and the RING domain family. The HECT domain plays a role mainly through the sulfur ester bond necessary for the formation of the catalytic effect with ubiquitin, while the RING domain provides a residential site for E2 and substrate to enable E2 to catalyze the transfer of ubiquitin to the substrate. Recently, a new E3 family, the U-box protein family, was discovered. The characteristic of the U-box E3 ligase is that it has a conserved U-box domain of approximately 70 amino acids. Secondary structure prediction showed that the U-box was a modified RING domain-missing scaffold and conserved zinc-chelated cysteine and histidine residues in a typical RING domain. There are 64 U-box genes in *Arabidopsis* and 77 in rice. These genes participate in biotic and abiotic stress processes, fertility and hormone signaling pathways [4], indicating that the plant U-box (Plant U-box, PUB) protein family regulates many processes of plant growth and development.

*PUB* genes play essential roles in plant resistance to disease. For example, in *Arabidopsis*, NPR1 is ubiquitinated and involved in salicylic acid (SA)-mediated plant immune responses to bacterial and fungal pathogens [5,6]. *Arabidopsis* ubiquitin protein AtPUB4 plays a vital role in cell proliferation of the shoot and apical meristem [7,8,9]. Moreover, the E3 ubiquitin ligases AtPUB12, AtPUB13, AtPUB22, AtPUB23 and AtPUB24 are well-studied negative regulators of *Arabidopsis* immunity that enhance host resistance to various pathogens in mutants. AtPUB12 and AtPUB13 polyubiquitinate the bacterial PAMP flg22 and promote flagellin-induced FLS2 degradation, and the mutant *pub12/13* displayed elevated immune responses to flagellin treatment [10,11,12]. The homologous triplets PUB22, PUB23, and PUB24, which respond to several distinct immune responses, including the oxidative burst, MPK3 activity, and transcriptional activation of marker genes, were increased and/or prolonged. Enhanced activation of PTI responses also resulted in increased resistance against bacterial and oomycete pathogens, accompanied by increased production of reactive oxygen species and cell death [13,14]. SPL11 (OsPUB11), the U-box protein of *Oryza sativa*, has broad-spectrum resistance to rice pathogens, increases defense gene expression and reactive oxygen species levels. Also, it is involved in flowering time control in rice through ubiquitination of different target proteins [15,16,17,18,19]. In addition, Wang revealed the regulatory mechanism of PUB25 and PUB26 polyubiquitinated BIK1 in *Arabidopsis* [20]. Murata et al. showed that AtCHIP could improve the adaptability of *Arabidopsis* under stress conditions [21]. Yang et al. found that AtPUB17 participates in the ETI pathway and HR, and regulates the cell death process [22]. Kobayashi suggested that the expression of AtPUB20 was induced by PAMPs and had autoubiquitination activity *in vitro*. The substrate of AtPUB20 is AGB1, which localized in the nucleus and cytoplasm [23]. Wang et al. found that OsPUB75 participated in cell division and elongation by mediating the BR signaling pathway [24]. Ishikawa clarified that OsPUB44 in rice might promote the virulence of pathogens through specific interactions with XopP [25]. Zhou et al. reported that NtCMPG1 regulates different aspects of PTI by working with members of group III E2 precisely [26]. He et al. found that NbPUB17 located in the nucleus and promoted specific immune pathways triggered by *Phytophthora infestans* [27]. Bos et al. found that PAMP-like elicitin INF1-induced cell death (ICD) was inhibited by blocking the autoubiquitination of StCMPG1 [28]. Soybean cyst nematode (SCN, *Heterodera glycines*) is one of the most important plant-parasitic nematodes, which is a major threat in all soybean production regions and causes USD 1.2 billion and USD 120 million in annual yield losses in the United States and China, respectively [29,30,31]. Several studies showed that the effector proteins secreted by SCN and some resistance genes of soybean play crucial roles in the interaction between soybean and SCN [32,33,34,35,36]. However, the function of soybean ubiquitination genes (*GmPUB* genes) in soybean–SCN interaction is largely unknown.

Since *PUB* genes play a crucial role in multiple plants’ immunity against a large variety of pathogens, we believe that the *GmPUB* genes may influence responses to SCN infection and involve some regulatory pathways. Toward this end, the response and expression patterns of 49 *GmPUB* genes under SCN infection were investigated in this study. Then, we overexpressed and silenced these *GmPUB* genes, which strongly responded to SCN infection in transgenic soybean hairy roots, and investigated the development of SCN. This study demonstrated the changes and functional roles of *GmPUB* genes under SCN infection, which provided an in-depth understanding of the molecular mechanism of the interaction between soybean and soybean cyst nematode.

## 2. Results

### 2.1. Bioinformatics Analysis of GmPUBs

The U-box domain is highly conserved in eukaryotes. In the present study, multiple sequence alignment showed that 49 GmPUB proteins contained a complete and conserved U-box domain (Figure 1). At the same time, according to Wang’s method [37], we analyzed the domains of 49 GmPUBs, and these GmPUBs were divided into six categories in the present study. The C-terminus of the U-box domain of type II PUB has an ARM repeat structure, and 21 of 49 soybean PUBs belong to this type. The domain of the type III PUB contained the GKL isomerase domain, including 5 soybean PUBs. Class IV PUBs contain kinase domains, and 5 of the 49 soybean PUBs are classified as such. Class V PUBs contain only U-box domains, 16 of 49 soybean PUBs. Type VI and type VII PUBs contained only one protein. The C-terminus of the type VI PUB contains the WD40 domain, and type the VII PUB contains the TPR structure (Appendix A, Figure 2).

Then, the physicochemical properties of the amino acid sequences of 49 GmPUB family members were analyzed using the online tool Protparam of ExPASy Proteomics. The results show that the relative molecular weights of GmPUBs were concentrated in 23.65–166.32 kD (Table 1); the highest theoretical isoelectric point was *Glyma12g188900* (9.11), and the lowest was *GlymaU008400* (4.7).

Furthermore, to compare the genetic relationships among the 49 *GmPUB* genes, the amino acid sequences of 49 GmPUB proteins were downloaded from NCBI and used for constructing a phylogenetic tree (Figure 2).

### 2.2. Expression Pattern Analysis of GmPUB Genes under Nematode Stress

The qRT–PCR results demonstrated that all 49 *GmPUB* genes changed at four time points, but the expression levels were different (Figure 3). The expression level of six genes changed significantly: *Glyma01g131800*, *Glyma07g106000*, *Glyma03g088400*, *Glyma15g038600*, *Glyma02g195900* and *Glyma12g188900*. At the same time, according to the relationship between the proteins encoded by these genes and PUB proteins in *Arabidopsis*, we renamed them *GmPUB16A*, *GmPUB20A*, *GmCHIPA*, *GmPUB33A*, *GmPUB23A* and *GmPUB24A* (Appendix A). In HPZ, four genes (*GmPUB16A*, *GmCHIPA*, *GmPUB23A* and *GmPUB24A*) showed a trend of first decreasing and then increasing, and the relative expression levels were the lowest at 10 dpi. The expression level of *GmPUB20A* increased with inoculation time. The relative expression of the *GmPUB33A* gene first decreased and then increased. In Williams 82 (W82), the relative gene expression of *GmPUB16A* and *GmPUB24A* after SCN infection increased first and then decreased and peaked at 5 dpi. *GmCHIPA* showed a trend of first decreasing and then increasing. *GmPUB23A* showed an upward trend after longer induction times, while *GmPUB20A* and *GmPUB33A* showed a downward relative gene expression trend (Figure 4).

### 2.3. Cloning of Five GmPUB Genes

Specific primers were designed according to the full-length sequence, and PCR was performed to obtain *GmPUB* genes. The length of the CDS region was as follows: *GmPUB20A*: 1314 bp; *GmCHIPA*: 837 bp; *GmPUB33A*: 2454 bp; *GmPUB23A*: 1257 bp; and *GmPUB24A*: 1284 bp. The electrophoresis and sequencing results show that five full-length sequences of *GmPUBs* were successfully obtained. The *GmPUB16A* was not successfully cloned in different soybean cultivars after several times (Appendix A).

### 2.4. Multiple Sequence Alignment of GmPUBs

Using the amino acid sequence of the *GmPUB* gene as a probe, several similar proteins in *Arabidopsis thaliana*, *Theobroma cacao*, *Capsicum annuum* and *Oryza sativa* were found in the GenBank database, indicating that GmPUB proteins are ubiquitous in higher plants. DNAMAN software was used to align the sequences of the GmPUB proteins with AtPUB23 (*At2g35930*), AtPUB22 (*At3g52450*) and AtPUB24 (*At3g11840*). The amino acid sequences of AtPUB25 (*At3g19380*), AtPUB20 (*At1g66160*), AtPUB18 (*At1g10560*), AtPUB19 (*At1g60190*), TcPUB23 (*Thecc1E022966*), CaPUB1 (*ABA59556*) and OsPUB15 (*Os08g01900*) were subjected to multiple alignment (Appendix A), and it was found that the amino acid sequences of proteins coded by the *GmPUB* genes were highly similar to those of PUB proteins in other species. This suggests that their function is highly conserved.

### 2.5. Phylogenetic Analysis of GmPUBs

The amino acid sequences of PUB proteins from different plants were downloaded from NCBI, and the phylogenetic analysis of GmPUBs (Figure 5) showed that six GmPUB proteins (GmPUB16A, GmPUB20A, GmCHIPA, GmPUB33A, GmPUB23A and GmPUB24A) showed relatively high similarities with four *Arabidopsis* PUB proteins (AtPUB18, AtPUB19, AtPUB22 and AtPUB23). Among them, GmPUB16A, GmPUB20A, GmPUB23A and GmPUB24A were clustered in the same large branch, whereas other two GmPUB proteins (GmCHIPA and GmPUB33A) were clustered in the other two branches. On the branches of GmPUB16A, GmPUB20A, GmPUB23A and GmPUB24A, four GmPUB proteins were closely related to AtPUB16, AtPUB20, TcPUB23 and AtPUB24, clustered in the same branch, and evolved in four different directions. In GmCHIPA and GmPUB33A, the former has high homology with AtCHIP, while the latter has high homology with JrPUB33.

### 2.6. Construction of Vectors for Overexpressing and Silencing GmPUBs

Specific primers were used for PCR amplification of the recombinant overexpressing vectors, and 1% agarose gel electrophoresis was performed to check the size of PCR products. The sizes of PCR products were as follows: pOE-GmPUB20A: 1593 bp; pOE-GmCHIPA: 1116 bp; pOE-GmPUB33A: 2733 bp; pOE-GmPUB23A: 1536 bp; and pOE-GmPUB24A: 1563 bp (Appendix A). This indicated that five plant recombinant vectors overexpressing the *GmPUB* were successfully constructed.

The RNA interference vectors pRNAi-GmPUB16A, pRNAi-GmPUB20A, pRNAi-GmPUB24A, pRNAi-GmPUB33A, pRNAi-GmCHIPA and pRNAi-GmPUB23A were digested with restriction enzymes. Electrophoresis showed three bands (2094 bp, 2627 bp, 8837 bp), three bands (2094 bp, 2624 bp, 8834 bp), three bands (2094 bp, 2634 bp, 8834 bp), three bands (2094 bp, 2557 bp, 8767 bp), two bands (1542 bp, 12,951 bp) and two bands (1558 bp, 11,998 bp), respectively, in line with the expectation (Appendix A), indicating that six interference vectors were successfully constructed.

### 2.7. Gene Expression and Biomass Analysis of Hairy Roots

The positive soybean hairy roots were detected and screened with a handheld lamp 3415RG (Luyor, Shanghai, China) to visualize GFP expression. Hairy roots with strong GFP fluorescence signals were reserved and used for further testing, and the non-GFP hairy roots and the main roots were removed (Appendix A).

qRT–PCR was used to verify the expression level of the *GmPUB* gene in the hairy roots overexpressing and RNA interfering with the *GmPUB* gene. The outcomes displayed that the copy number of the *GmPUB* gene in the hairy roots overexpressed the *GmPUB* gene more than the control plant (EV). The pOE-GmPUB20A and pRNAi-GmCHIPA were upregulated by 472.50 and 150.44 times, respectively. Accordingly, the copy number of the *GmPUB* gene in hairy roots that interfered with the expression of the gene was lower than that in the control plant (EV), and the downregulation multiple was between 0.14–0.67. Among them, the most obvious downregulation was observed for *GmPUB20A* and *GmPUB23A*. In addition, compared with transgenic soybean hairy roots harboring empty vectors, the number of lateral roots of hairy roots overexpressing the *GmPU*B gene increased significantly, while the number of lateral roots of hairy roots RNA interfering with the expression of the *GmPUB* gene decreased significantly (Figure 6 and Figure 7).

### 2.8. GmPUBs Regulate Soybean Resistance to SCN in Hairy Roots

After the hairy roots were inoculated with SCN J2, the samples were collected at 15 dpi, and the nematodes in soybean roots were stained by the sodium hypochlorite-acid fuchsin method. The nematodes in each stage were examined and counted under stereomicroscope. As shown in Figure 8, it was found that the total number of nematodes in soybean hairy roots overexpressing *GmPUB20A*, *GmPUB33A* and *GmPUB24A* was higher than EV, while the total number of nematodes in soybean hairy roots overexpressing *GmCHIPA* and *GmPUB23A* was lower than that in EV. The results show that overexpression of *GmPUB20A*, *GmPUB33A* and *GmPUB24A* could weaken the resistance of soybean plants under nematode stress, while overexpression of *GmCHIPA* and *GmPUB23A* could enhance the resistance of soybean to SCN.

A further study of the resistance of soybean hairy roots with *GmPUB* RNA interfered with SCN, as shown in Figure 9, compared with EV; the total number of nematodes in soybean hairy roots after RNA interference with *GmPUB* genes decreased significantly. Among them, the total number of nematodes in soybean hairy roots after interference to *GmCHIPA* and *GmPUB23A* decreased significantly. The above results showed that soybean plants that interfered with the expression of the *GmPUB* gene had strong resistance.

## 3. Discussion

Ubiquitin ligase E3 is an important factor that determines substrate specificity in the ubiquitin protein degradation pathway. The plant U-box (PUB) protein family contains the U-box domain, which is a class of ubiquitin ligase E3 with important functions [38,39,40] and plays a wide regulatory role in plant growth and development, biological and abiotic stress, fertility and hormone signaling pathways [4]. There are few reports about soybean *GmPUB* responses to nematode stress. In the process of soybean and SCN interaction, the relationship between *GmPUB* resistance to nematode stress and its dependent signaling pathways is unclear.

Recently, 125 U-box genes were identified in the soybean genome [37]. Gene expression analyses can provide critical information about the potential functions of *GmPUB* genes [41]. Researchers analyzed the expression profile of *GmPUB* genes under SCN stress using publicly available RNA-seq datasets [42]. Based on the analysis of the RNA-seq datasets, 36 soybean *GmPUB* genes were found to have significantly changed after incubation with SCN. These results support the notion that *GmPUB* likely plays a vital role in soybean immunity against SCN [41]. Accordingly, this study was based on publicly available RNA-seq data and previous transcriptome data from our research group, and six soybean *GmPUB* genes (*GmPUB16A*, *GmPUB20A*, *GmCHIPA*, *GmPUB33A*, *GmPUB23A* and *GmPUB24A*) were screened from 49 *GmPUB* genes by qRT–PCR. The differential gene expression of these six genes at different time points after SCN infection indicates that they may be related to the interaction of soybean and SCN. Meanwhile, the public RNA-seq data investigated the response of soybean roots to virulent and avirulent SCN inoculation for 6 and 8 days, while this study examined the response of genes in susceptible and resistant soybeans at four time points (1, 5, 10 and 15 dpi), which expanded our understanding of the functions of ubiquitination-related genes in soybean immunity. Most soybean cultivars resistant to SCN are derived from limited resistance sources, and SCN races have begun evolving to overcome SCN resistance [41,43]. Therefore, studying the function of these six genes in the process is necessary.

In plants, U-box is often coupled with ARM (armadillo repeat) to form a U-box/ARM structure, which interacts independently or jointly with other proteins. The U-box/ARM protein is a special protein in higher plants [44]. There were 41 U-box/ARM proteins in *Arabidopsis* and 43 U-box/ARM proteins in rice, but there was no U-box/ARM protein in *Chlamydomonas vulgaris* [45]. Some proteins with U-box/ARM functional domains have been identified in other higher plants. These proteins were the ARC1 protein in rape [46], ACRE276 protein in tobacco [22] and PUB4 protein [47], PHOR1 protein in potato [48], CMPG1 protein in celery (*Petroselinum crispum*) [49], BG55 protein in mangrove plant *Bruguiera gymnorrhiza* [50] and SPL11 protein in rice [16]. Some scholars believe that the U-box/ARM protein plays an important role in the plant hormone response, disease resistance and other aspects [51,52]. Among these proteins, U-box proteins such as SPL11 in rice [18,19,53], ACER276 in tobacco and AtPUB17 in *Arabidopsis* are related to plant disease resistance [22,27]. SPL11 is involved in the primary defense of rice against pathogen infection and negatively regulates cell death [16,19]. Contrary to the negative regulation of SPL11 on disease resistance, CMPG1 in tomato, tobacco and celery can positively regulate plant disease resistance [22]. Meanwhile, Amador et al. reported that PHOR1 (photoperiod-responsive1), a U-box gene found in *Solanum tuberosum*, is a positive regulator of the GA signal transduction pathway [48]. Kim et al. found that NtPUB4 in tobacco is involved in plant development and cytokinin signal transduction [47]. However, studies on the disease resistance of the U-box/ARM protein in soybean have not been reported. In the present study, 49 soybean *GmPUB* genes differentially expressed under nematode stress were analyzed by bioinformatics, and 49 U-box proteins were divided into six categories. The U-box/Arm structure is the second type, and 21 of the 49 soybean PUBs belong to this type of protein. The *GmPUB16A* (*Glyma01g131800*) gene was highly expressed at different time points after inoculation with SCN. The silencing of *GmPUB16A* in hairy roots decreased the number of SCN. It indicates that the U-box/Arm structure proteins may play a regulatory role in plant disease resistance. These results provide a theoretical basis and reference for further studying the mechanism of U-box proteins in soybean and cyst nematode interaction.

The CHIP protein located in the cytoplasm, with auxiliary chaperone molecules and E3 activities. Its E3 activity depends on the C-terminal U-box domain [53]. Through its E3 activity and molecular chaperones, such as Hsc70/Hsp70 and Hsp90, CHIP synergistically changes the Hsc70/Hsp70- and Hsp90-mediated regulation of signaling pathways in the protein folding and degradation balance and the quality control of proteins in cells [21,54]. The *Arabidopsis* AtCHIP has three peptide repeats and one U-box domain. Its expression is induced by various environmental stresses, such as the increase in expression under the stimulation of high temperature, low temperature or strong light and the overlapping repair or degradation of these abnormal proteins by regulating Hsp70 and Hsp90 to reduce the damage of environmental stress on plants [44,55]). Other studies have found that the AtCHIP protein can promote the degradation of abnormal proteins in cells under strong light stress using the subunit ClpP4 of chloroplast protease [56]. The AtCHIP protein is also related to the signal transduction pathway of abscisic acid (ABA) stress, and plants overexpressing the *AtCHIP* gene are very sensitive to ABA stress [57]. The AtCHIP protein has become a typical example for studying the physiological functions and mechanisms of the AtPUB protein. In this study, GmCHIPA (encoded by *Glyma03g088400*) was highly homologous to AtCHIP. In addition, the *GmCHIPA* gene was expressed at different time points after inoculation with SCN J2. A further study of its function showed that overexpression of *GmCHIPA* reduced the number of nematodes. Silencing this gene also causes a decrease in the number of nematodes. However, this research has not yet clarified the specific mechanism by which *GmCHIPA* regulates soybean cyst nematode resistance. We hypothesized that the overexpression and silencing of *GmCHIPA* are involved in different regulatory pathways between soybean and cyst nematode interaction. Therefore, understanding the specific substrates of GmCHIPA and the upstream and downstream signaling pathways involved needs to be further investigated.

As positive or negative regulators of the plant immune response to various pathogens, U-box proteins play an important role in the plant defense response and programmed cell death [39]. Many types of U-box proteins play a negative regulatory role in plant disease resistance. The U-box E3 gene *Spl11* was isolated from the indica rice variety IR68, and its mutant *spl11* could increase the nonspecific resistance of rice to *Magnaporthe grisea* and *Xanthomonas oryzae pv. Oryzae* [16]. Similarly, Liu et al. reported that the U-box E3 ligase SPL11 and its *Arabidopsis* homologous gene *PUB13* in rice played a negative regulatory role in PCD and defense responses [19]. *Arabidopsis* U-box E3 ubiquitin ligases AtPUB22, AtPUB23 and AtPUB24 negatively regulate the PTI response induced by PAMPs such as elf18 and chitin. Studies have shown that the PTI response was significantly enhanced in the three mutants of *pub22**/pub23**/pub24*, and the three mutants significantly improved the resistance to live vegetative pathogens. At the same time, *Pseudomonas syringae pv.* tomato inoculated with *pub22/pub23/pub24* mutants showed enhanced resistance to pathogens, increased ROS content and upregulated expression of *RbohD* genes involved in ROS formation [13,14,58]. In this paper, GmPUB23A (encoded by *Glyma02g195900*) and GmPUB24A (encoded by *Glyma12g188900*) are homologous to AtPUB23 and AtPUB24. Functional studies have found that the overexpression of *GmPUB24A* weakened the resistance of soybean under nematode stress, and the overexpression of *GmPUB23A* enhances the resistance of soybean to SCN. Furthermore, this study interfered with the *GmPUB24A* and *GmPUB23A* in soybean hairy roots. The results show that soybean roots interfered with the expression of the *GmPUB24A* and *GmPUB23A* genes, conferring strong resistance to SCN. Similarly, in *GmCHIPA*, soybean’s resistance to SCN was enhanced in both *GmPUB23A* overexpression and RNA interference hair roots. We speculate that the reason for this situation is that plants under two treatments may also be involved in different regulatory pathways of resistance to nematodes. In addition, Wang et al. showed that the soybean *GmPUB8* gene, which was highly homologous to *GmPUB23A*, played an important role in abiotic stress. Its expression was induced by exogenous ABA and NaCl, regulating flowering time through the photoperiod pathway and playing a negative regulatory role under plant drought stress [37]. Similarly, the tolerance of *Arabidopsis pub22* and *pub23* mutants to salt stress was enhanced, while *pub22pub23* double mutants showed an additive effect [59].

Unlike the above U-box proteins that play a negative regulatory role, some plant U-box proteins play a positive regulatory role in plant disease resistance. AtPUB17, AtPUB20 (AtCMPG1) and MAC are positive regulatory factors in plant disease resistance, and these three proteins are involved in the regulation of ETI in the innate immune response. Among them, the expression of AtPUB20 was induced by PAMP and had autoubiquitination activity *in vitro*. The substrate of AtPUB20 is the G-protein β subunit AGB1 [23,60]. To regulate host immunity, soybean GmPUB1-1 interacts with many other RxLR effectors, including Avr1d [61]. Lin et al. confirmed that GmPUB13 is involved in the virulence mechanism of *Phytophthora effectors*. The virulence mechanism of GmPUB13 is that Avr1d inhibits the activity of the GmPUB13 E3 ligase by competing with E2 to stabilize the sensitive factor GmPUB13 and promote *Phytophthora* infection [11]. In this study, GmPUB20A (encoded by *Glyma07g106000*) was homologous to AtPUB20 (AtCMPG1), and its function was further studied. The overexpression of *GmPUB20A* under nematode stress weakened the resistance of plants. In contrast, interfering with the expression of *GmPUB20A* more significantly enhanced the resistance of soybean to cyst nematodes. These findings highlight the involvement of U-box proteins in plant immune responses. In addition, the transmission of defense signals and the participation of UPS (ubiquitin-26S proteasome system) components in the process from pathogen perception to pathogen immune response and effector immune response also proved the multiple roles of UPS in biological stress response.

Although many ubiquitin ligases have been identified, their function is still unclear. The mining and identifying of more upstream regulatory factors and downstream substrates of ubiquitin ligases are essential for understanding their functions. Currently, studies on the function of U-box genes mainly use yeast two-hybrid screening of downstream proteins interacting with them [62] and reverse genetics based on RNA silencing or gene knockout [7]. In addition, the same U-box ubiquitin ligase may involve various biological processes. For example, the U-box ubiquitin ligases AtPUB20 and AtPUB23 are involved in plant disease resistance, growth and development. The AtCHIP protein is related to both temperature stress and abscisic acid stress. This action mode of ubiquitin ligase also provides a new basis for studying some cross-signaling pathways. Although research on PUB is a great challenge for plant science, it also helps us to understand the protein–protein interaction in plants and many physiological activities in cells, especially the function of such proteins in plant disease resistance and stress resistance. *GmPUB20A*, *GmPUB33A* and *GmPUB24A* played a vital role in soybean resistance to cyst nematodes in the present study, and several problems need to be solved immediately, such as finding the substrates of *GmPUB* genes and clarifying the process of ubiquitin proteins’ recognizing substrate proteins. The results of these future studies will be more helpful in explaining the mechanism of plants against pathogen infection through immune defense and provide new ideas for the integrated management of nematode disease.

## 4. Materials and Methods

### 4.1. Nematode Population, Plant Materials and Growth Condition

The race 3 (HG 0) population of *Heterodera glycines* (soybean cyst nematode, SCN) was propagated on the susceptible soybean cultivar Liaodou 15. Soybean (*Glycine max*), Williams 82 (W82) and “Huipizhi” (HPZ) were selected as susceptible and resistant cultivars to SCN race 3, respectively. Soybean seeds were surface sterilized with 1% NaClO for 10 min and were washed at least five times with sterilized water. Then, the seeds were germinated in plastic pots (length × width × height = 22 × 15 × 7 cm) containing wetting vermiculite with Hoagland nutrient solution. Four days later, uniform seedlings were selected and transferred into PVC tubes (height × diameter = 10 × 3 cm) containing equal ratios of sterilized sand and soil in a climatic chamber (16 h light/8 h dark, temperature 23–26 °C, humidity 50%). The plants were watered once every three days with Hoagland nutrient solution.

### 4.2. Bioinformatics Analysis of GmPUBs

The accession number and coding sequence (CDS) of soybean *GmPUB* genes were obtained from SoyBase (https://www.soybase.org, accessed on 1 October 2020), and the molecular weight and isoelectric point (PI) of the gene product were analyzed online by Protparam (http://web.expasy.org/protparam/, accessed on 4 April 2021). The protein sequences of the *GmPUB* genes were downloaded from the NCBI website. DNAMAN software was used for multisequence alignment, and the NJ algorithm of MEGA 7.0 was used to generate the phylogenetic tree.

### 4.3. Expression Analysis of GmPUBs

Quantitative real-time PCR (qRT–PCR) was performed to detect the relative expression of the 49 *GmPUB* genes of the susceptible cultivar William 82, and the resistant variety Huipizhi after soybean cyst nematode race 3 infection in this study (approximately 2000 second-stage juveniles (J2) were inoculated at the root of each soybean seedling, and uninoculated plants were used as controls). SCN-infected soybean roots were collected at 1 dpi, 5 dpi, 10 dpi and 15 dpi. RNA was extracted from soybean roots using the Reagent RNA extraction kit (CWbio, Beijing, China) according to the manufacturer’s instructions. Then, RNA was used for first-strand cDNA synthesis by using the PrimerScript^TM^ RT Reagent Kit with gDNA Eraser (TaKaRa Bio Inc., Dalian, China). qRT–PCR was performed by using a TaKaRa SYBR Green PCR Master Mix kit, and the qRT–PCR system and procedures were performed according to the manufacturer’s instructions. All qRT–PCR experiments were performed on a CFX Connect Real-Time Quantitative PCR system (Bio-Rad, Hercules, CA, USA) with the manufacturer’s instructions. The soybean ubiquitin 3 gene *GmUBI-3* (GenBank accession D28123.1) was used as an internal reference gene, with three parallel and three biological repeats per sample. The results were analyzed by the 2^–ΔΔCT^ method, and all the specific primers used for qRT–PCR are shown in Appendix A.

### 4.4. Cloning of 6 GmPUB Genes

cDNA of soybean Williams 82 was used for gene cloning. First, RNA was extracted from soybean roots by using the Reagent RNA extraction kit (CWbio, Beijing, China) according to the manufacturer’s instructions. Then, RNA was used for first-strand cDNA synthesis (unified concentration of 1000 ng μL^−1^) using PrimerScriptTM RT Reagent Kit with gDNA Eraser (TaKaRa Bio Inc., Dalian, China). The full-length primers CDS-F and CDS-R were designed according to the 6 *GmPUBs* (Appendix A). The full-length sequences of these *GmPUBs* were obtained by amplifying leaf cDNA as a template.

### 4.5. Construction of Vectors for Overexpressing and Silencing GmPUBs

The plant expression vector pNINC2RNAi with a GFP tag was constructed using pCAMBIA3301 as the backbone [63]. Primers were designed according to the cDNA sequence of soybean *GmPUBs* and the sequence of AscI and AvrII restriction sites in pNINC2RNAi and homologous recombination requirements (Appendix A). Then, the full-length CDS of 5 *GmPUBs* was cloned into pNINC2RNAi via PCR technology.

For *GmPUB16A*, *GmPUB20A*, *GmCHIPA*, *GmPUB33A*, *GmPUB23A* and *GmPUB24A* RNAi, we synthesized the RNAi sequences (including sense sequence, loop and antisense sequence) in BioRun Biosciences Co., Ltd. (BioRun, Wuhan, China). Then, the synthetic fragment was inserted into the vector pNINC2RNAi by restriction digest cloning to obtain the 6-plant interference recombinant vector pNINC2RNAi-RNAiGmPUBs of the *GmPUB* gene.

After screening positive clones, the freeze–thaw method transformed the recombinant vectors’ overexpressing and silencing *GmPUBs* into *Agrobacterium rhizogenes* K599.

### 4.6. Transformation Process of Soybean Hairy Roots

The seeds of soybean Williams 82 were disinfected and germinated in sterilized vermiculite watered regularly with Hoagland nutrient solution. Six days later, young seedlings with unexpanded cotyledons were used for infection with *A. rhizogenes* K599. The soybean cotyledonary nodes were inoculated with K599 strains carrying cognate plasmids, pNINC2RNAi-OEGmPUBs, pNINC2RNAi (empty vector, EV) and pNINC2RNAi-RNAiGmPUBs, for overexpression or silencing by syringe [64]. After hairy roots were grown, the following parts of the hairy roots were subtracted (Appendix A). The plants with hairy roots were transferred into a new plastic culture pot containing sterile vermiculite for further culture, and nutrient solution and sterile water were regularly irrigated. Approximately ten days later, the hairy roots were screened with a handheld lamp (Luyor, Shanghai, China) to visualize *GFP* expression. Hairy roots carrying strong *GFP* signals were reserved and transferred into plastic culture pots for further culture.

### 4.7. Gene Expression and Biomass Analysis of Hairy Roots

The samples were collected from hairy roots cultured for 15 d, and the effects of overexpression and silencing were confirmed by detecting the expression level of the *GmPUB* gene by qRT–PCR (the primers used for qRT–PCR are shown in the Appendix A). The soybean ubiquitin 3 gene *GmUBI-3* (GenBank accession D28123.1) was used as an internal reference gene, with three parallel and three biological repeats per sample, and the results were analyzed by the 2^–ΔΔCT^ method. At the same time, the biomass of hairy roots cultured for 15 d was indicated by the number of lateral roots of soybean-transformed hairy roots. Each treatment was repeated five times.

### 4.8. Investigation of SCN Development in GmPUB-Overexpressing and -Silenced Soybean Hairy Roots

When the hairy root length was approximately 5–10 cm, it was transferred into PVC tubes (height × diameter = 10 × 3 cm) containing equal ratios of sterilized sand and soil in a climatic chamber (16 h light/8 h dark, temperature 23–26 °C, humidity 50%). The plants were watered once every three days with Hoagland nutrient solution. After the hairy roots were restored to health, the second-stage juveniles (J2) of SCN were inoculated into the hairy roots induced by pNINC2RNAi-OEGmPUBs, pNINC2RNAi and pNINC2RNAi-RNAiGmPUBs. Each plant was inoculated with 500 J2s, and hairy roots without nematodes were used as a control. The nematodes in soybean roots were stained by the sodium hypochlorite-acid fuchsin method at 15 dpi with SCN. The nematodes at different ages were examined and counted under an anatomical microscope and microscope. Each treatment was repeated five times.

### 4.9. Statistical Analyses

Statistical analyses were conducted using IBM SPSS STATISTIC v.22 (Armonk, NY, USA) and GraphPad Prism 9 software (GraphPad Inc., San Diego, CA, USA). Independent *t*-tests were applied in SPSS to analyze the difference in the number of nematodes and lateral roots. Multiple *t*-tests were applied to detect the significant differences in the relative expression levels of all *GmPUBs* in the pairs HC vs. HS, WC vs. WS, EV vs. OE and EV vs. RNAi on GraphPad Prism 9, and the Holm–Sidak method was used to determine the statistical significance with alpha = 0.05.

## 5. Conclusions

This study’s results clearly indicate the characteristics of 49 *GmPUB* genes in soybean and their expression patterns under SCN stress. Six soybean *GmPUB* genes with significant responses to SCN infection were obtained. Additional functional studies of the six genes showed that *GmPUB**20A*, *GmPUB**33A* and *GmPUB24A* negatively regulated soybean resistance. These data help to further reveal the regulatory mechanism of soybean *GmPUB* participating in soybean-SCN resistance.

## Figures and Tables

**Figure 1 ijms-23-10771-f001:**
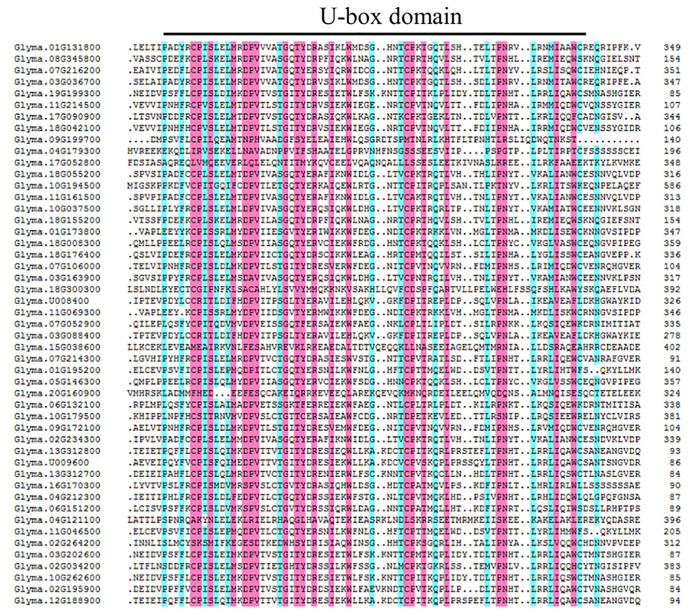
Multiple alignment of amino acid sequences of 49 GmPUBs proteins. Note: pink and blue indicate 75% and 50% homology of amino acids, respectively. The number on the right of each amino acid sequence represents the relative position of the last residue. The U-box domain is marked at the top of the alignment with black solid lines.

**Figure 2 ijms-23-10771-f002:**
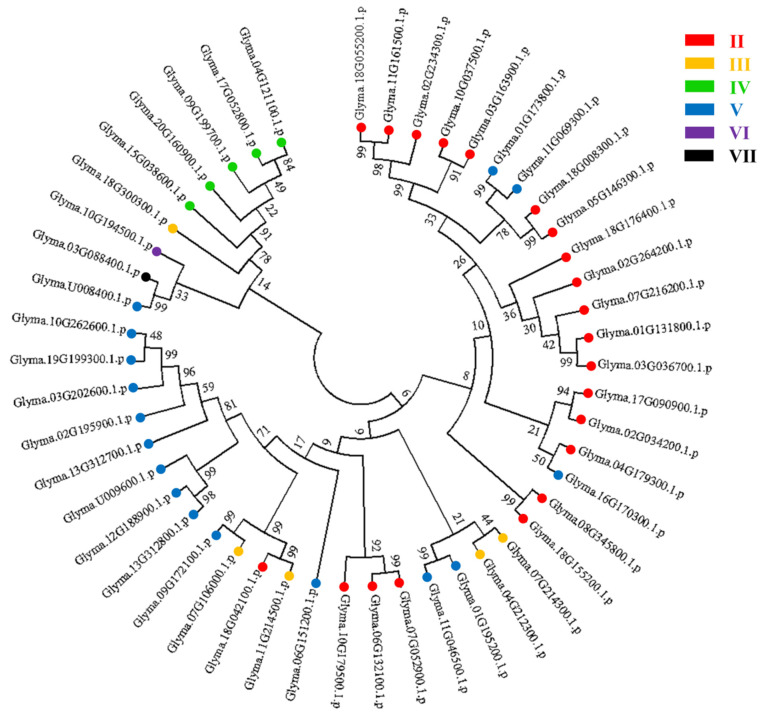
The phylogenetic tree of 49 GmPUB proteins. Note: the neighbor-joining (NJ) tree was constructed based on the full-length protein sequences of 49 soybean *GmPUB genes*.

**Figure 3 ijms-23-10771-f003:**
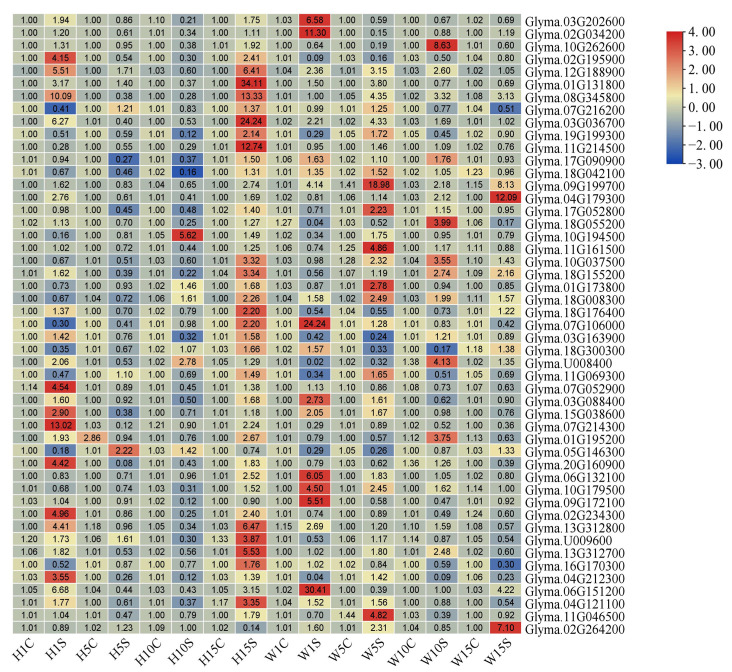
The heatmap of *GmPUBs’* expression level of susceptible cultivar Williams 82 and resistant cultivar HPZ during soybean cyst nematode infection. Note: H1C and W1C represent the expressions of *GmPUBs* without soybean cyst nematode infect at 1 dpi, respectively. H1S and W1S represent the relative expressions of *GmPUBs* after soybean cyst nematode infection at 1 dpi, respectively. Likewise, the samples of H5C, W5C, H5S and W5S were taken at 5 dpi. The samples of H10C, W10C, H10S and W10S were taken at 10 dpi. Additionally, the samples of H15C, W15C, H15S and W15S were taken at 15 dpi. The relative expression value of *GmPUBs* higher than 1 and lower than 1 represent upregulation and downregulation, respectively.

**Figure 4 ijms-23-10771-f004:**
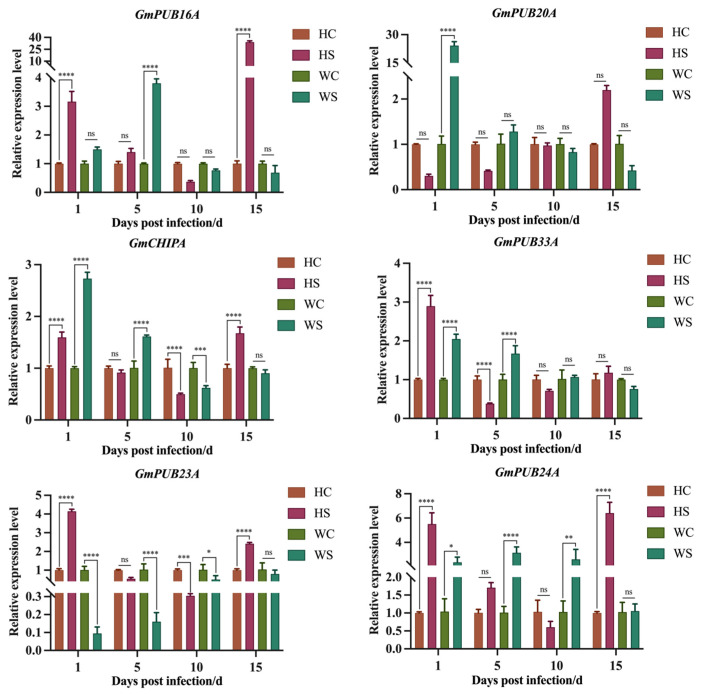
The response of the six *GmPUBs* of susceptible cultivar Williams 82 and resistant cultivar HPZ during soybean cyst nematode infection. Note: control: HC and WC represent the relative expressions of *GmPUBs* without soybean cyst nematode infect, respectively. Treatment: HS and WS represent the relative expressions of *GmPUBs* during soybean cyst nematode infect, respectively. The statistical comparisons were conducted with the *t*-test for members of treatment and control. ns represent not significant. *, **, *** and **** indicate significant difference at the *p* < 0.05, *p* < 0.01, *p* < 0.001 and *p* < 0.0001 probability levels, respectively.

**Figure 5 ijms-23-10771-f005:**
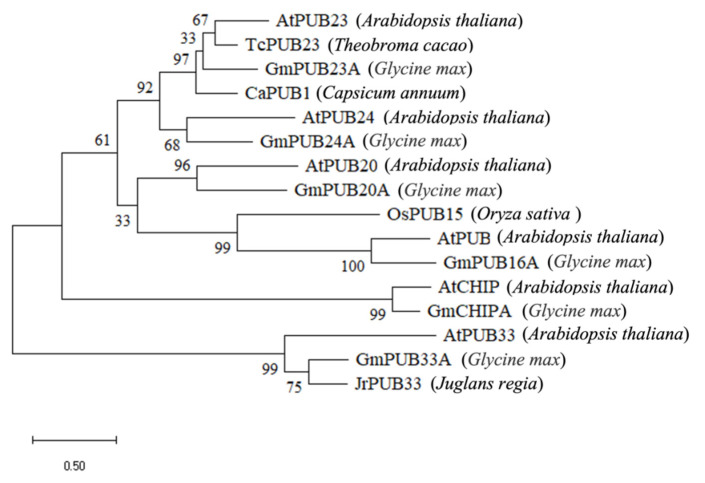
Phylogenetic analysis of six GmPUB proteins.

**Figure 6 ijms-23-10771-f006:**
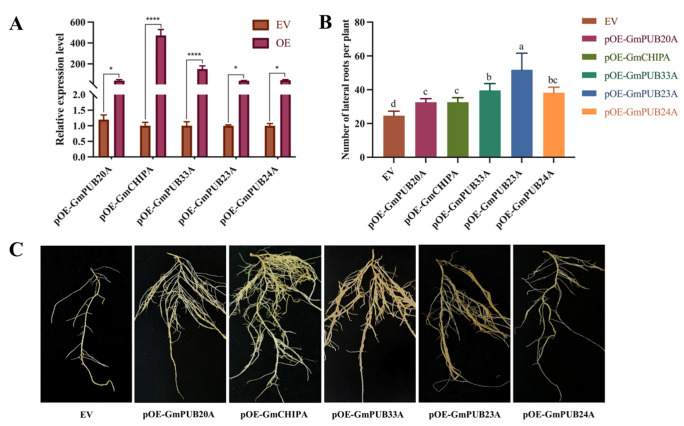
Expression profiles and biomass analysis of overexpression of five *GmPUBs* in the hairy roots. Note: (**A**), expression profiles of five *GmPUBs* overexpressed hairy roots, (**B**), biomass analysis of *GmPUBs* overexpressed hairy roots, (**C**), lateral roots of the *GmPUBs* overexpressed hairy roots. One-way ANOVA was applied in SPSS to analyze the difference of lateral roots in EV and transgenic hairy roots. Multiple *t*-tests were applied to detect the significant differences of the relative expression level of all *GmPUBs* in EV and other transgenic hairy roots on Graphpad prism 9. ns represent not significant. * and **** indicate significant difference at the *p* < 0.05 and *p* < 0.0001 probability level, respectively. Different characters mean significant differences found at *p* < 0.05 level.

**Figure 7 ijms-23-10771-f007:**
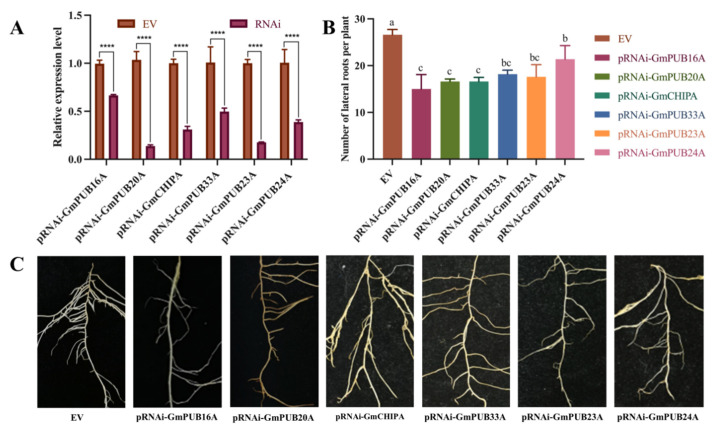
Expression profiles and biomass analysis of six *GmPUBs* in the RNAi hairy roots. Note: (**A**), The relative expression of six *GmPUBs* in the RNAi hairy roots, (**B**), Biomass analysis of *GmPUBs* RNAi hairy roots, (**C**), Lateral roots of *GmPUBs* RNAi hairy roots. One-way ANOVA was applied in SPSS to analyse the difference the number of lateral roots in EV and transgenic hairy roots. Multiple *t*-tests were applied to detect the significant differences of the relative expression level of all *GmPUBs* in EV and other transgenic hairy roots on Graphpad prism 9. ns represent not significant. **** indicate significant difference at the *p* < 0.0001 probability level. Different characters mean significant differences found at *p* < 0.05 level.

**Figure 8 ijms-23-10771-f008:**
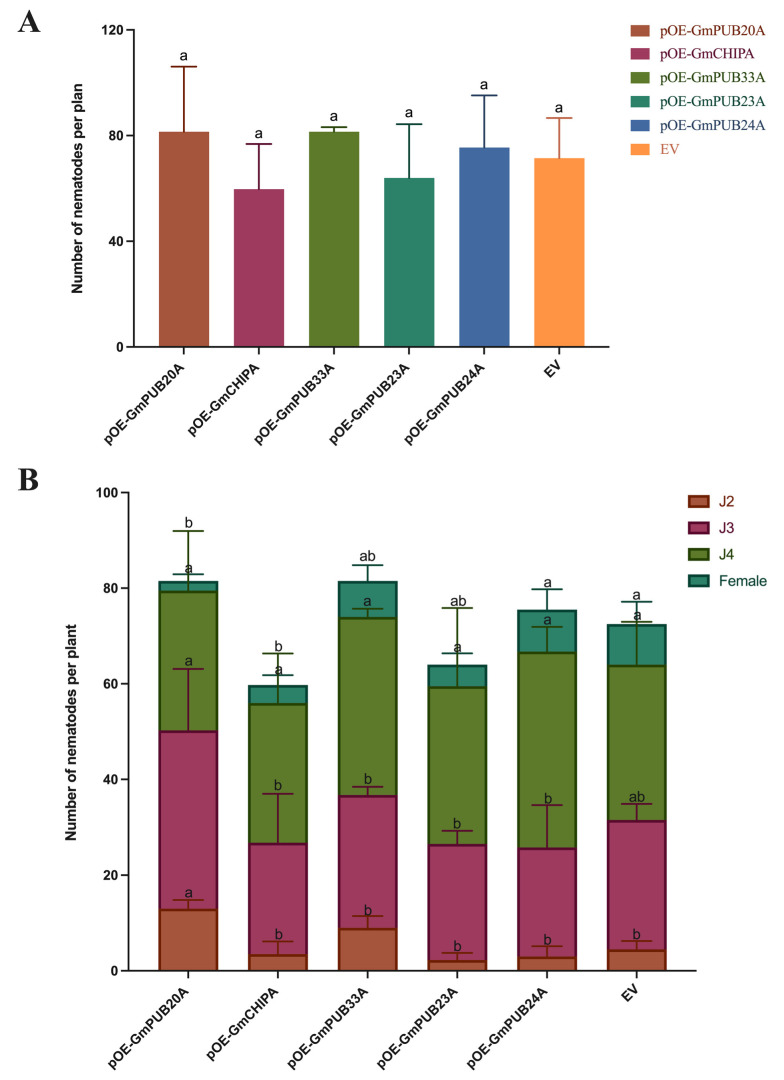
Number of nematodes in the five *GmPUBs*-overexpressed hairy roots. Note: (**A**), total number of nematodes per plant, (**B**), number of different stage nematodes per plant. One-way ANOVAs were applied in SPSS to analyze the difference in the number of nematodes in EV and other transgenic hairy roots, and the Holm–Sidak method was used to determine the statistical significance. Different characters mean significant differences found at *p* < 0.05 level.

**Figure 9 ijms-23-10771-f009:**
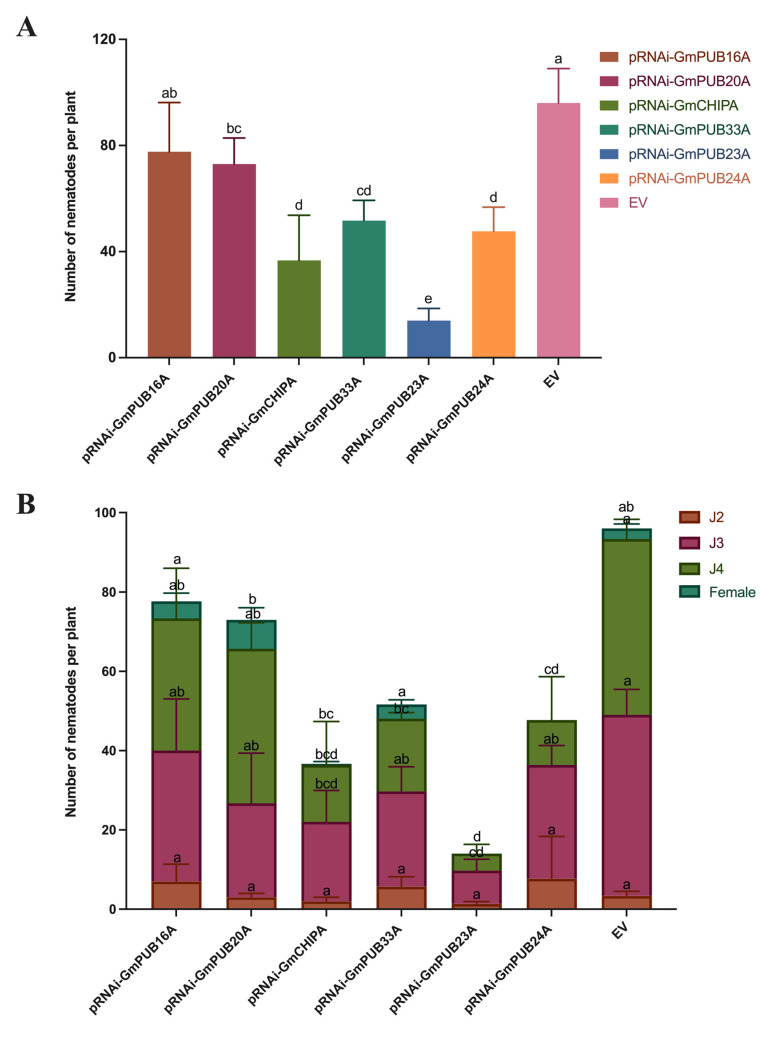
Number of nematodes in the six *GmPUBs* RNAi hairy roots. Note: (**A**), total number of nematodes per plant, (**B**), number of different stage nematodes per plant. One-way ANOVAs were applied in SPSS to analyze the difference in the number of nematodes in EV and other transgenic hairy roots, and the Holm–Sidak method was used to determine the statistics. Different characters mean significant differences found at *p* < 0.05 level.

**Table 1 ijms-23-10771-t001:** The physical and chemical characteristics of amino acids of PUBs from soybean.

ID	Protein Length (aa)	Molecular Weight (Da)	Theoretical pI
*Glyma01g131800*	702	76,147.81	7.12
*Glyma06g317700*	807	89,644.06	7.14
*Glyma08g345800*	461	51,427.67	8.76
*Glyma07g216200*	654	71,514.08	7.13
*Glyma03g036700*	488	53,949.81	5.57
*Glyma19g199300*	419	46,769.44	8.84
*Glyma11g214500*	435	48,243.16	7.8
*Glyma17g090900*	676	74,288.16	8.88
*Glyma18g042100*	431	47,580.43	7.41
*Glyma09g199700*	800	91,665.81	6.16
*Glyma04g179300*	525	57,492.61	6.69
*Glyma17g052800*	760	85,495.34	5.61
*Glyma18g055200*	841	91,179.81	5.6
*Glyma10g194500*	1481	166,316.78	5.49
*Glyma11g161500*	838	90,638.29	5.55
*Glyma10g037500*	663	74,369.1	5.65
*Glyma18g155200*	461	51,280.61	8.75
*Glyma01g173800*	762	85,859.94	6.23
*Glyma18g008300*	768	85,120.94	6.11
*Glyma18g176400*	617	67,784.83	4.99
*Glyma07g106000*	437	49,082.93	8.47
*Glyma03g163900*	856	94,847.76	7.53
*Glyma18g300300*	1089	122,436.52	6.29
*GlymaU008400*	207	23,646.88	4.7
*Glyma11g069300*	761	85,318.37	5.98
*Glyma07g052900*	1006	112,155.25	5.69
*Glyma03g088400*	278	31,631.97	5.25
*Glyma15g038600*	817	91,927.31	6.5
*Glyma07g214300*	414	45,136.29	7.11
*Glyma01g195200*	449	49,000.23	5.73
*Glyma05g146300*	764	84,505.61	5.15
*Glyma20g160900*	738	83993.49	5.56
*Glyma06g132100*	1007	112,597.01	6.57
*Glyma10g179500*	1014	112,389.33	5.5
*Glyma09g172100*	438	49,150.02	7.48
*Glyma02g234300*	813	88,446.46	5.48
*Glyma13g312800*	422	47,506.5	8.75
*GlymaU009600*	417	46,784.76	9.02
*Glyma13g312700*	403	45,151.96	8.94
*Glyma16g170300*	414	44,685.01	8.61
*Glyma04g212300*	393	44,075.61	7.17
*Glyma06g151200*	417	45,549.14	8.68
*Glyma04g121100*	810	90,253.97	7.02
*Glyma11g046500*	514	56,194.04	6.19
*Glyma02g264200*	682	74,758.43	6.28
*Glyma03g202600*	420	46,866.57	8.78
*Glyma02g034200*	713	79,524.84	8.38
*Glyma10g262600*	419	46,727.2	8.67
*Glyma02g195900*	418	45,950.59	7.99
*Glyma12g188900*	427	48,136.26	9.11

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
