# Peer review of "Functional Characterization of Ubiquitination Genes in the Interaction of Soybean—Heterodera glycines"

_ijms, 2022, doi:10.3390/ijms231810771_

Round 1

Reviewer 2 Report

Dear Authors,

I have an opportunity to review paper entitled:” Functional Characterization of Ubiquitination Genes in the Interaction of Soybean-Heterodera glycines” submitted to the section ‘Molecular Plant Sciences’ in IJMS journal.

Authors concentrated on the response of soybean GmPUB genes to soybean cyst nematode (SCN, Heterodera glycines) infection;

I suggest only to present the research hypothesis before between the introduction and results section; Moreover, I encourage to add information about the investigated soybean-Heterodera glycines interaction what it has already known in this pathosystem interaction ? in current form the reader was founded only information about yield losses;

It is also quite strange when Authors used term “PUB genes responded to SCN infection”- please correct to more appropriated form;

Furthermore, how do we know that “Williams 82 (W82) and “Huipizhi” (HPZ) were selected as susceptible and resistant cultivars to SCN”

The research design appropriate and material and methods quite clear described;

Results are in general quite clear, but the qRT-PCR should be done in comparison to the two reference genes ! One reference gene is too weak;

Please fit the Figure 3 to the whole A4 page to make the obtain results more visible; the same situation with figure 4; it is important findings in relative genes expression -confirmation of transcriptional results;

Discussion is interesting and conclusion quite short and firm, moreover, Authors finished this part with the statement “Although 416 research on PUB is a great challenge for plant science, it also helps us to understand the 417 protein-protein interaction in plants and many physiological activities in cells”- therefore, please, provide some future prospects coming from obtained results to make the findings interesting to the wider audience in plant-pathogen pathosystems;

Additionally, Line 93 -please correct the text;

Author Response

Response to Reviewer 2 Comments

Firstly, thanks for your valuable suggestions to our manuscript! We have updated our manuscript according to your suggestions, please see the latest version.

Point 1: I suggest only to present the research hypothesis before between the introduction and results section; Moreover, I encourage to add information about the investigated soybean-Heterodera glycines interaction what it has already known in this pathosystem interaction ? in current form the reader was founded only information about yield losses.

Response 1: We haved added the research hypothesis before between the introduction and results section. Moreover, the information about the investigated soybean-Heterodera glycines interaction has been added in our manuscript. Please see Line 87-100.

Point 2: It is also quite strange when Authors used term “PUB genes responded to SCN infection”- please correct to more appropriated form.

Response 2: We have replaced “PUB genes” with “GmPUB genes” to make it more appropriate.

Point 3: How do we know that “Williams 82 (W82) and “Huipizhi” (HPZ) were selected as susceptible and resistant cultivars to SCN”.

Response 3: Williams 82 is a susceptible cultivar to SCN, and it also has a high-quality genome annotation. Therefor, Williams 82 is often selected as a susceptibile host in soybean-SCN interaction (Lewers, et al., 2001; Noel, et al., 2003). Moreover, Huipizhi is a elite resistant cultivar to several SCN races (race 1, 3, 4, 5, 7, and 14), so it was selected as resistant cultivar in our research (Bin et al., 2014; Li et al., 2018). Please see the followed references:

Lewers, K.S.; Nilmalgoda, S.D.; Warner, A.L.; Knap, H.T.; Matthews, B.F. Physical mapping of resistant and susceptible soybean genomes near the soybean cyst nematode resistance gene Rhg4. Genome 2001, 44, 1057-64.

Noel, G.R.; Wax, L.M. Population Dynamics of Heterodera glycines in Conventional Tillage and No-Tillage Soybean/Corn Cropping Systems. Journal of nematology 2003, 35, 104-109.

Li, S.; Chen, Y.; Zhu, X.; Wang, Y.; Jung, K.-H.; Chen, L.; Xuan, Y.; Duan, Y. The transcriptomic changes of Huipizhi Heidou (Glycine max), a nematode-resistant black soybean during Heterodera glycines race 3 infection. Journal of Plant Physiology 2018, 220, 96-104.

Bin, L.; Sun, M.J.; Lan, W.; Zhao J.R.; Wang, Z.L. Comparative analysis of gene expression profiling between resistant and susceptible varieties infected with soybean cyst nematode race 4 in Glycine max. Journal of Integrative Agriculture 2014, 13, 2594-2607.

Point 4: Results are in general quite clear, but the qRT-PCR should be done in comparison to the two reference genes ! One reference gene is too weak.

Response 4: For all qRT-PCR experiments in this study, we performed pre-experiment to compared the stability in the expression of four candidate reference genes including the GmUBI-3, ACT11, CYP2 and ELF1A. Finally, we found that the expression of GmUBI-3 was most stable in non-SCN infected and SCN infected soybean roots.

Point 5: Please fit the Figure 3 to the whole A4 page to make the obtain results more visible; the same situation with figure 4; it is important findings in relative genes expression -confirmation of transcriptional results.

Response 5: We have updated the format of Figure 3 and Figure 4 to fit the A4 page, please the new version in our manuscript.

Point 6: Discussion is interesting and conclusion quite short and firm, moreover, Authors finished this part with the statement “Although 416 research on PUB is a great challenge for plant science, it also helps us to understand the 417 protein-protein interaction in plants and many physiological activities in cells”- therefore, please, provide some future prospects coming from obtained results to make the findings interesting to the wider audience in plant-pathogen pathosystems.

Response 6: We have added the future prospects in the ending of the last paragraph of the Discussion section. Please see Line 744-750.

Point 7: Additionally, Line 93 -please correct the text.

Response 7: We have corrected the mistakes in our manuscript.

Other changes:

We have modified our manuscript according to suggestions of reviewer #1. We also correct the grammer or spelling mistakes in our manuscript. All the changes were labbeled in red color. Please see the latest version.

Round 2

Reviewer 2 Report

In my opinion Authors correct manuscript according most of suggestions;

However in my opinion more information about Heterodera glycines should as a pathogen be added in introduction;

Especially hypothesis, conclusions or results presentation form was significantly improved;

Moreover, Authors added response as a covincing explanation